# Comprehensive Carbon Emission and Economic Analysis on Nearly Zero-Energy Buildings in Different Regions of China

Yiting Kang [1,2], Jianlin Wu [2], Shilei Lu [1,*], Yashuai Yang [2], Zhen Yu [2], Haizhu Zhou [2], Shangqun Xie [2], Zheng Fu [3], Minchao Fan [1] and Xiaolong Xu [4]

1   School of Environment Science and Engineering, Tianjin University, Tianjin 300072, China
2   China Academy of Building Research, Beijing 100013, China
3   School of Energy and Environmental Engineering, Hebei University of Technology, Tianjin 300401, China
4   China Association of Building Energy Efficiency, Beijing 100029, China
*   Correspondence: lvshilei@tju.edu.cn

**Abstract:** Considering the comprehensive effect of building carbon emissions, cost savings is of great significance in nearly-zero-energy buildings (NZEBs). Previous research mostly focused on studying the impact of technical measures in pilot projects. The characteristics of different cities or climate zones have only been considered in a few studies, and the selection of cities is often limited. At times, only one city is considered in each climate zone. Therefore, this study selected 15 cities to better cover climate zone characteristics according to the variation in weather and solar radiation conditions. A pilot NZEB project was chosen as the research subject, in which the energy consumption was monitored and compared across different categories using simulated values by EnergyPlus software. Various NZEB technologies were considered, such as the high-performance building envelope, the fresh air heat recovery unit (FAHRU), demand-controlled ventilation (DCV), a high-efficiency HVAC and lighting system, daylighting, and photovoltaic (PV). The simulated carbon emission intensities in severe cold, cold, and hot summer and cold winter (HSCW) climate zones were 21.97 $kgCO_2/m^2$, 19.60 $kgCO_2/m^2$, and 15.40 $kgCO_2/m^2$, respectively. The combined use of various NZEB technologies resulted in incremental costs of 998.86 $CNY/m^2$, 870.61 $CNY/m^2$, and 656.58 $CNY/m^2$. The results indicated that the HSCW region had the best carbon emission reduction potential and cost-effectiveness when adopting NZEB strategies. Although the incremental cost of passive strategies produced by the envelope system is higher than active strategies produced by the HVAC system and lighting system, the effect of reducing the building's heating load is a primary and urgent concern. The findings may provide a reference for similar buildings in different climate zones worldwide.

**Keywords:** sensitivity analysis; incremental cost; energy simulation; climate regions





## 1. Introduction

In order to limit the rise in atmospheric temperature to 2 °C, or even 1.5 °C, the Paris Agreement calls on all parties to develop long-term low-emission development strategies based on their national conditions and capabilities [1]. The Intergovernmental Panel on Climate Change (IPCC) in 2018 announced that the impact of extreme harm from greenhouse gas emissions can only be avoided if the world achieves net zero greenhouse gas emissions by the middle of the 21st century [2]. Developed and developing countries have submitted declarations to achieve this carbon neutrality target over the next 30–60 years [3–7]. In many countries, building energy consumption accounts for 40% of the total energy consumption [8]. In order to reduce operational carbon emissions in the building sector, a series of policies on nearly-zero-energy building (NZEB) have been implemented, and the development of NZEB could be a mid-to-long-term strategy [9,10]. The roadmap of NZEB involves reducing buildings' energy consumption and maximizing renewable energy usage to achieve the core purpose of reducing primary energy

consumption [11–13]. The demonstrated NZEBs, technical guides, and policies [14] have promoted the development of NZEBs in developed countries. According to the Asia-Pacific Economic Cooperation (APEC) statistics data, China accelerated the building energy-saving work by promoting ultra-low-energy buildings and NZEBs, with the consequences of nearly 10 million square meters by 2020 [14]. However, the uncertainty of comprehensive and systematic performance cost strategies in the scope of different climate zones hinders the promotion of NZEBs.

It is well known that adopting combined and optimized technical strategies for reducing building energy consumption and using renewable energy are the main pathways for achieving NZEBs [15–19]. The energy-saving effect of single or multiple common NZEB technical strategies has been widely demonstrated and analyzed by energy consumption software in previous studies [15,16,20]. Strategies can be classified into four categories during the design stage [17,21]: (1) passive design and building envelope; (2) high-performance heating, ventilation, and air conditioning (HVAC) systems; (3) high-efficiency lighting systems; and (4) renewable energy utilization.

Optimizing the performance of the building envelope in terms of improving thermal performance to reduce the heating load in winter and decreasing the shading coefficient (SC) to reduce the cooling load in summer is critical in the design stage of passive design and building envelopes. Heating load could be significantly decreased by increasing the insulation thickness in severe cold and cold zones in China [22]. However, in the HSCW zone of China, a related study [23] showed that the decrease in the operational energy demand in NZEB is lower when using a thick insulation layer, and the unfavorable highly insulated envelope may even cause a risk of overheating in summer. Chen [24] has demonstrated that the high SC value shading system has an obvious energy saving effect based on a parameterized building energy consumption assessment model in the HSCW zone.

High-efficiency HVAC equipment, such as air-source heat pumps and ground-source heat pumps, are widely applied in NZEBs according to the cases analyzed and results from the U.S. and China [25]. Air-source heat pumps are considered a primary energy alternative and are frequently used to achieve electrification. The operational performance of a super-low-temperature combined solar air collector, air-source heat pump, and energy-storage hybrid system has been proven in an ultra-low-energy building in a severely cold region of China. Furthermore, to forecast HVAC systems' energy demand, a novel hybrid modelling structure was contrived [26], and reinforcement learning was considered for HVAC systems in an intelligent building [27]. A fresh-air heat recovery unit (FAHRU) was adopted in NZEBs in a fresh-air supply system using Chinese and EU standards [17,28], and previous research shows that a combined heat recovery and photovoltaic (PV) unit could result in a yearly energy consumption reduction of up to 67.5% with a payback period of 3.5 years in hot and cold regions [28]. Previous study [29] illustrated the effect of shading and daylighting performance on energy savings and economic feasibility, mostly benefitting cooling load reduction. As for PV application installed on building roofs and facades of ZEBs, Liu et al. [30] studied the load-matching issue by selecting one typical city in each of China's climate zones. An analysis from PV calculation software shows that the orientation, installation angle, and solar radiation are the main influencing factors in renewable energy generation [31].

The studies mentioned above have focused on studying the impact of technical measures in pilot projects. Only in a few cases were different cities or climate zones considered, and the selection of cities was often limited, at times only assessing one city per climate zone. Xing Su [23] analyzed the energy consumption and carbon emissions of an office building by comparing the baseline building and passive building, but only in one city in the HSCW zone of China. Chen [24] have studied the energy saving effect of shading system only in HSCW zone of China. S.D. de Garayo [27] had optimized a combined HVAC system in a passive house in hot and cold regions. Liu et.al [30] illustrated the effect of PV by only choosing five cities to represent the whole climate zones of China. Furthermore,

research on specific technical measures of NZEB are limited to the feasibility study on technical effects [32] and ignoring comprehensive analysis on technical sensitivity and economic analysis in different climate regions.

Therefore, nearly zero-energy building technologies should be illustrated in whole climate zones. With the well-perceived variation in weather and solar radiation conditions, inclusion of represented typical cities in the study would lead to better coverage of climate zone features. Hence, this paper takes a typical pilot NZEB project as the research subject, verifies the simulated energy consumption with the test data, then extends the analysis of carbon emission reduction rate and incremental cost to 15 cities across severe cold, cold and HSCW climate zone of China. Technical sensitivity analysis has been performed to discover carbon emission reduction potential by altering parameters of envelope heat transfer co-efficiency and shading co-efficiency (SC) value, coefficient of performance (COP) of HVAC system, adopting strategies of demand control ventilation (DCV) and FAHRU. The discovery of this paper might provide a valuable reference for similar climate zones around the world.

## 2. Methodology

### 2.1. Calculation Model and Assumption

#### 2.1.1. Simulation Model

To quantitatively investigate the energy saving impact on NZEB technologies, the simulation method is adopted to evaluate the energy consumption of different climatic regions of China. EnergyPlus software (EPS) is a widely acceptable energy simulation tool around the world. It was built on two existing programs in terms of DOE-2 and BLAST [33,34]. EPS could output the hourly energy consumption results by inputting user-specified construction, internal loads, schedules, and weather condition parameters. Additionally, EPS also calculated HVAC system timely by setting separate modules of external climate, the building geometry and construction, the air-conditioning system operating under the control system and the air distribution system. Further, PVSYST software developed by Swiss scientist Andre Mermaid & Co [35] is widely used [36,37] to design grid-connected, off-grid, and DC PV systems, as well as to estimate power generation and system optimization. Users can acquire simulation results by entering the location, weather conditions and material type of the PV system.

This paper presents analysis of combined energy consumption, power generation, and climate sensitivity for three critical climate zones, where NZEB strategies are applicable and adopted in engineering construction. Fifteen typical cities from a severe cold area, cold area, and hot summer, and cold winter area have been selected for the study. Figure 1 shows the climate zones and solar radiation distribution. Besides, weather data for simulation were provided by hourly Chinese Standard Weather Data [38,39].

#### 2.1.2. Energy Consumption Saving Model

The annual energy saving per unit area (ESPA) is defined as the evaluation index of building energy consumption, and the calculation formula is as follows:

$$\alpha = \frac{(EC_1 - EC_2)}{A} \tag{1}$$

where, $\alpha$ is annual energy saving per unit area, kWh/m$^2$; $EC_1$ is original energy consumption without energy saving technologies, kWh; $EC_2$ is energy consumption by using NZEB technologies, kWh; and $A$ is building floor area, m$^2$.

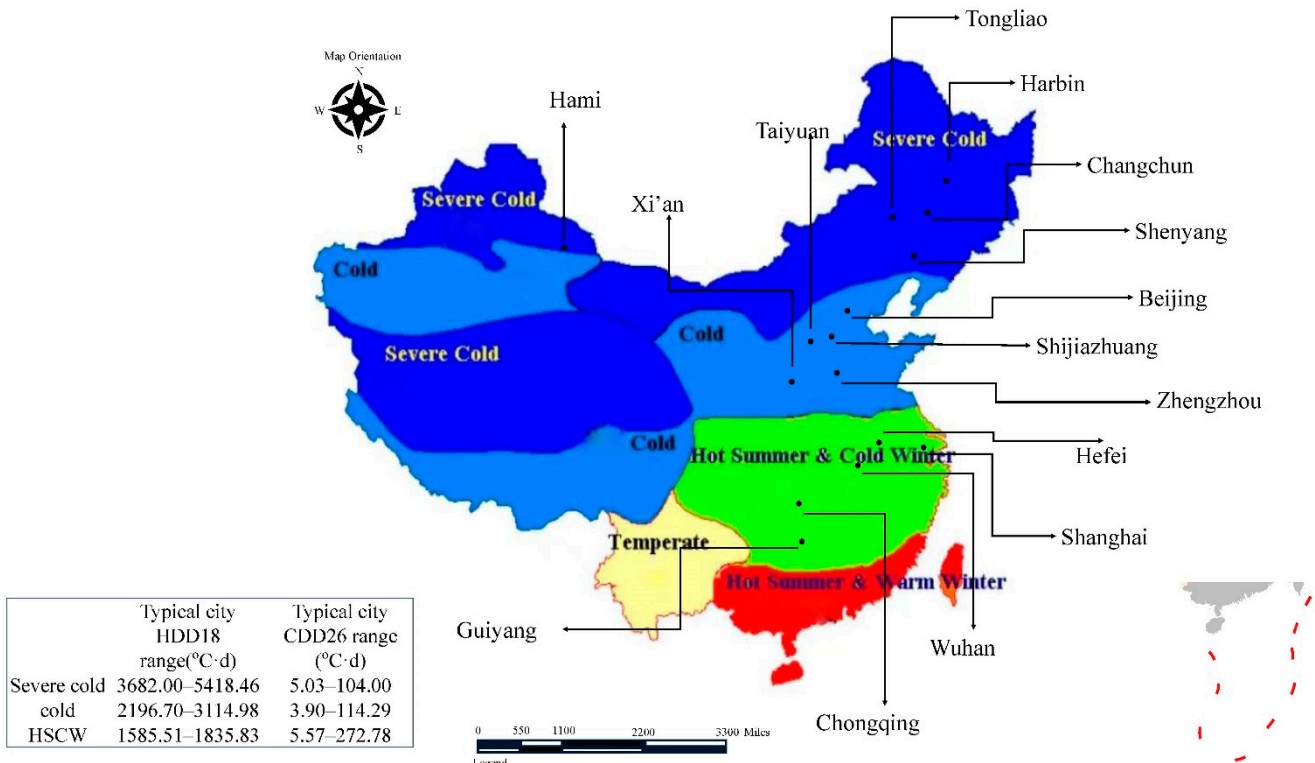

| | Typical city HDD18 range(°C·d) | Typical city CDD26 range (°C·d) |
|---|---|---|
| Severe cold | 3682.00–5418.46 | 5.03–104.00 |
| cold | 2196.70–3114.98 | 3.90–114.29 |
| HSCW | 1585.51–1835.83 | 5.57–272.78 |

(**a**) Typical cities in selected climate zones

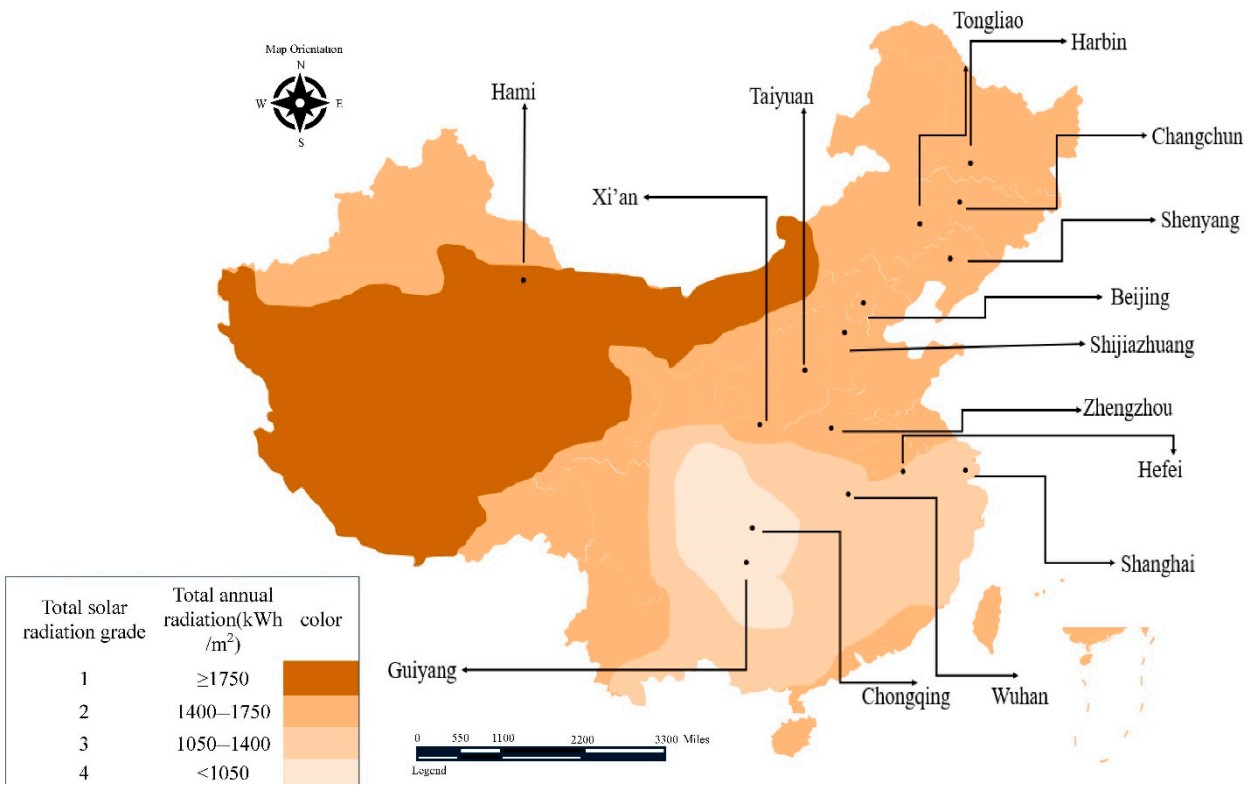

| Total solar radiation grade | Total annual radiation(kWh /m²) | color |
|---|---|---|
| 1 | ≥1750 | |
| 2 | 1400–1750 | |
| 3 | 1050–1400 | |
| 4 | <1050 | |

(**b**) Distribution of solar energy resources in China

**Figure 1.** Climate zones and solar energy resources of typical cities in China.

### 2.1.3. Carbon Emission Reduction Model

Based on the energy consumption model, carbon emission reduction model considers the unified regional electric carbon emission factor. Chinese Standard for Building Carbon Emission Calculation (GB/T 51366-2019 [40]) is referred for calculation of the operational carbon emission of buildings. The scope of carbon emission is categorized as: (i) building envelope system; (ii) HVAC system including heating and cooling source, FAHRU, and DCV; (iii) lighting fixture, lighting control and daylighting; and (iv) renewable energy system.

Formula (2) defines the calculation method of carbon emission, which is defined by GB/T 51366-2019 [40]. Carbon emission is calculated by energy consumption multiplying carbon emission factors.

$$C = EC \times EF_{grid} \tag{2}$$

where, $EC$ is energy consumption, KWh; $EF_{grid}$ is carbon emission factor of the grid in the local area, $kgCO_{2eq}/kWh$ [41].

Formula (3) and (4) define the annual carbon emission reduction per area and carbon emission reduction rate.

$$\beta = \frac{(CE_1 - CE_2)}{A} \tag{3}$$

$$R = \frac{(CE_1 - CE_2)}{CE_1} \tag{4}$$

where, $\beta$ is the annual carbon emission reduction per unit area, $kgCO_{2eq}/m^2$; $CE_1$ is original carbon emission without energy saving technologies, $tCO_2$; $CE_2$ is carbon emission by using NZEB technologies, $kgCO_{2eq}$; and $A$ is building floor area, $m^2$, $R$ is carbon emission reduction rate (%).

### 2.1.4. Incremental Cost Model

Formula (5) defines the annual incremental cost per unit area by using NZEB technologies.

$$\gamma = \frac{(C_1 - C_2)}{A} \tag{5}$$

where, $\gamma$ is annual incremental cost per unit area, $CNY/m^2$; $C_1$ is original cost without energy saving technologies, $CNY/m^2$; $C_2$ is cost by using NZEB technologies, $CNY/m^2$; and $A$ is building construction area, $m^2$.

### 2.2. Research Subject and Data Validation

### 2.2.1. Research Subject

This paper uses parameters of a typical nearly-zero-energy office building located in Beijing in the cold climate zone. To analyze the carbon emission of climate zones, thermal parameters for the reference building are listed in Table 1 based on the Chinese building energy saving standards [39,42]. The research subject is a commercial office building with 13,050 square meters, as illustrated in the simulation model in Figure 2a. To reduce the operational energy consumption and carbon emission, NZEB technologies were adopted. Tables 2 and 3 illustrate the setting condition of indoor parameter, energy-saving target and NZEB technology, respectively, for case building by using Chinese NZEB standard [17].

**Table 1.** Detailed input parameters of simulation model for baseline building scenario.

| Typical Areas | Envelope Details | | | | | Internal Load | | |
|---|---|---|---|---|---|---|---|---|
| | Heat Transfer Coefficient (W/m²·K) | | | Shading Coefficient (SC) | Window to Wall Ratio (WWR) | Occupancy Density (m²/Person) | Lighting Power Density (LPD) (W/m²) | Equipment Power Density (W/m²) |
| | Wall | Roof | Window | | | | | |
| Severe cold | 0.38 | 0.28 | 2.20 | 0.45 | 0.40 | 10 | 9 | 15 |
| Cold | 0.50 | 0.45 | 2.40 | 0.45 | 0.40 | 10 | 9 | 15 |
| HSCW | 0.60 | 0.4 | 2.50 | 0.45 | 0.40 | 10 | 9 | 15 |

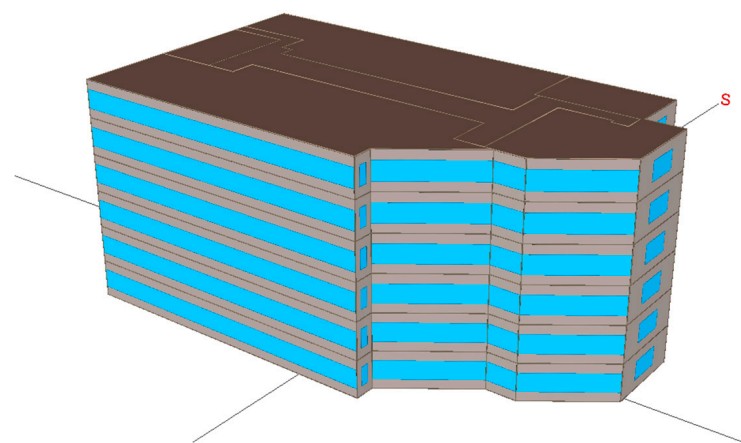

(**a**) The simulating 3D model of study case

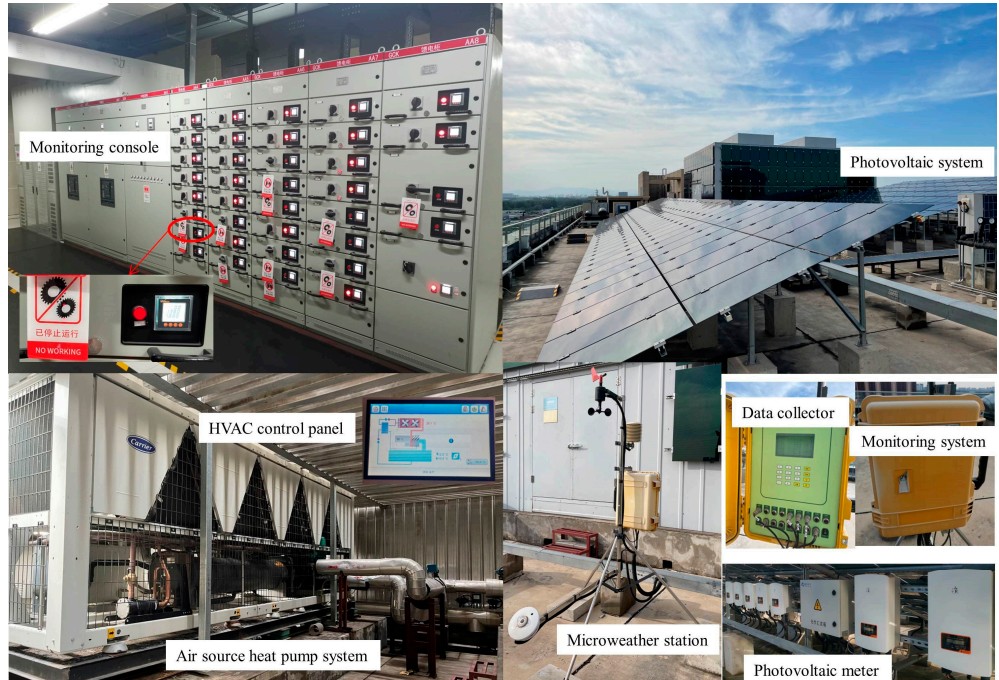

(**b**) Photos of adopted building energy saving strategies and energy management system

**Figure 2.** Model of case building and onsite photos.

**Table 2.** Indoor parameters and energy saving targets for research subject.

| Parameters | Winter | Summer |
|---|---|---|
| Temperature (°C) | 20 | 26 |
| Relative humidity (%) | 30 | 60 |
| Minimum fresh air volume [m$^3$/(h·per person)] | Office: 30, Hall: 10 | |
| Building energy saving rate (%) | 50% | |
| Building tightness | Air change times ($N_{50}$): 1.0 | |

**Table 3.** Detailed parameters for NZEB scenario of the research subject.

| Strategies | Building Parameters and Mechanical Systems | NZEB Technologies |
|---|---|---|
| Passive building strategies | Wall | Reinforced concrete wall with 50 mm rock wool and 30 mm vacuum insulation panel Heat transfer coefficient = 0.19 W/(m$^2$·K) |
| | Roof | Flat roof with 40 mm XPS and 30 mm vacuum insulation panel Heat transfer coefficient = 0.18 W/(m$^2$·K) |
| | Window | Triple glazing window with external shading Heat transfer coefficient = 1.0 W/(m$^2$·K) Shading coefficient (winter) = 0.47 Shading coefficient (summer) = 0.20 |
| Active building strategies | Space heating | Air source heat pump system (COP = 3.18) |
| | Space cooling | Air source heat pump system (COP = 2.60) |
| | Ventilation | Heat recovery fresh air system (enthalpy recovery efficiency = 75%) |
| | Domestic hot water system | Electrical water heater (thermal efficiency = 98%) |
| | Lighting system | High efficiency LED fixtures (LPD of office = 4.5 W/m$^2$; LPD of corridor = 2.0 W/m$^2$) |
| Renewable energy | Photovoltaics | PV system (installed capacity = 26 kWp) |

### 2.2.2. Model Validation

Energy management system has been established to monitor and record operational data of cooling source, heating source, air source heat pumps, water pumps, as illustrated in Figure 2b. For air condition system, separated sensors have been installed on air source heat pumps, circulating water pumps and fresh air system. Meters have been used for monitoring daily energy consumption and generation. To monitor the indoor environment, sensors are adopted to record temperature, relatively humidity, PM2.5 and $CO_2$ in typical room. Table 4 showcased the specific parameters of sensors. In this study, real-time operation data were used to verify the accuracy of the energy simulation model results.

**Table 4.** Characteristics of sensors.

| Name | Model | Accuracy | Measurement Range | Work Temperature |
|---|---|---|---|---|
| Electromagnetic flowmeter | AKE-CO3P | ±0.5% | ≤5 m/s | 25~60 °C |
| Temperature sensor | PT1000 | ±0.1 °C | 0~99.9 °C | 0~50 °C |
| Data recorder | Acrel DDSD1352 | 0.5 s Level | —— | 25~55 °C |

This paper presents a comparison of simulation results and monitoring results to validate the rationality of simulation method.

In order to validate the simulation accuracy of the models built by EPS, the monthly energy consumption of air source heat pumps and the annual energy consumption of water pumps and lighting are simulated and compared with the monitored data, respectively. Comparisons between the simulated and measured values are presented in Figures 3 and 4, respectively, which demonstrates excellent consistency. It is shown from the figures that the maximal difference between the monitored and simulated data of air source heat pumps is 8.3%, which is regarded as acceptable. For the water pumps and lighting system, the differences are 7.9% and 7.3%, respectively. Deviation method and results have been referred in similarly research [43,44]. The difference between air source heat pumps and water pumps is mainly caused by the weather data used in EPS. Hence,

the comparison results indicate that the energy simulation model is accurate and reasonable for the following analysis.

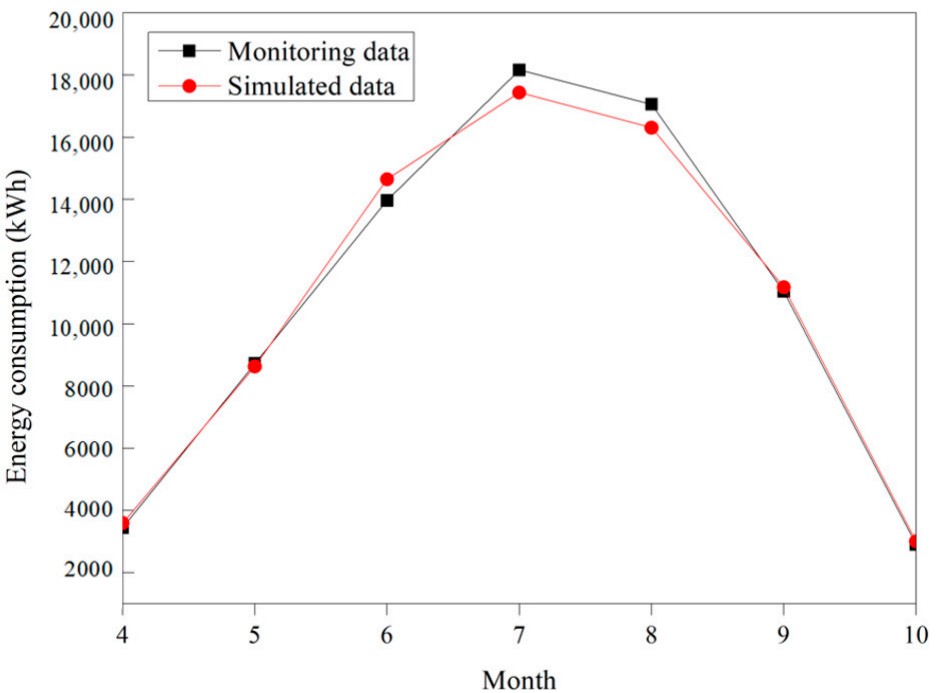

(**a**) Monthly energy consumption

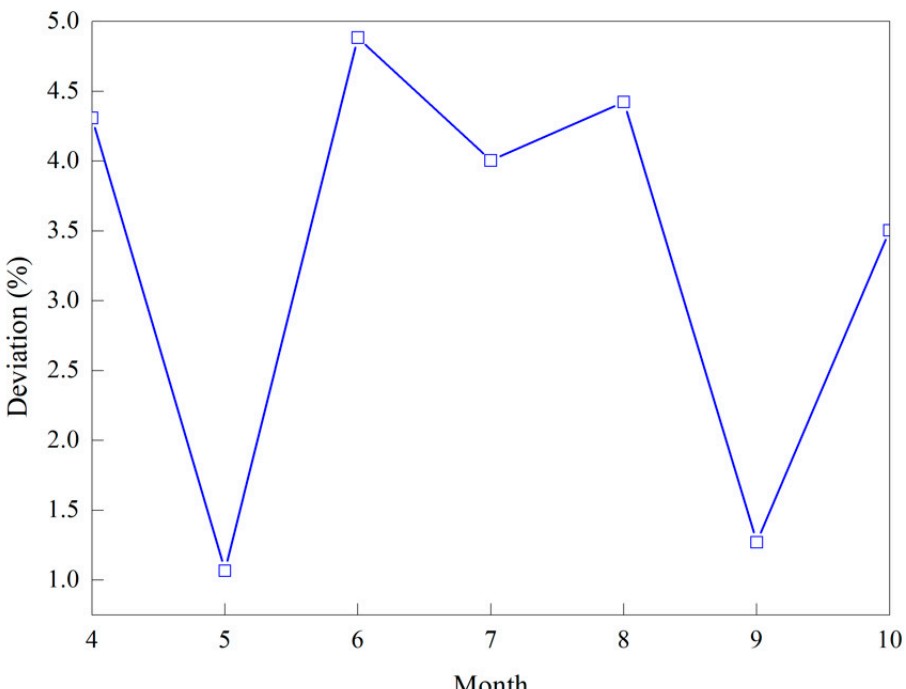

(**b**) Deviation between monitoring and simulated data

**Figure 3.** Monthly energy consumption comparison of air source heat pumps.

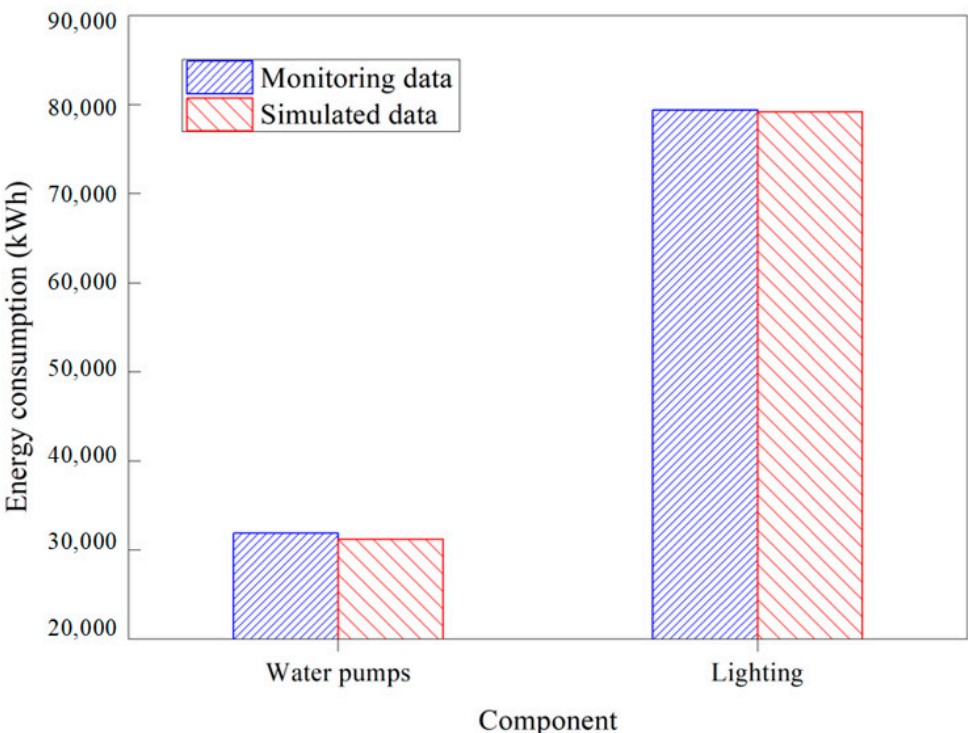

**Figure 4.** Annual energy consumption comparison of water pumps and lighting system.

## 3. Results and Discussions

### 3.1. Energy Saving Analysis of Different Technologies

This section discusses energy-saving technologies of NZEBs including high-performance envelopes, FAHRU, efficient HVAC and lighting systems, DCV, daylighting and renewable energy usage, etc. Climate discrepancy, as a key impact factor, is also considered in the analysis and discussion.

### 3.1.1. High-Performance Envelopes

Optimizing the performance of the building envelope, which is affected by the K and SC values, has a decisive impact on heating load and cooling load of the NZEB. Therefore, selecting high-performance envelope is an initial and critical step for NZEB. As shown in Figure 5, compared to the baseline, the average energy-saving per area (ESPA) of the buildings located in severe cold, cold, and HSCW climate zones is 6.63 kWh/m$^2$, 5.76 kWh/m$^2$, and 2.80 kWh/m$^2$, respectively, by adopting NZEB envelope parameters. The results show that the energy-saving potential of high-performance envelopes is superior to HSCW in the other two low-temperature regions due to the comprehensive effect of K value and SC value. Therefore, it is necessary to improve K value in severe cold and cold regions and enhance the SC value in HSCW area.

### 3.1.2. Heat Recovery Fresh Air System

In pursuit of a high-quality indoor air environment, sufficient filtered fresh air should be supplied in NZEBs. However, about 20–40% of the total energy consumption of air-conditioning system is consumed in the fresh air handling process [45,46]. It is critical to balance energy saving and indoor air quality. FAHRUs are common equipment in terms of latent heat recovery and total heat recovery, used in residential and commercial buildings. Annual energy savings caused by FAHRU mainly depend on the weather condition and fresh air volume. Annual ESPA changes with different weather conditions due to the fresh air volume are set as constant in this study. The red lines in Figure 6 plot the annual ESPA value of 4.00 kWh/m$^2$, 2.83 kWh/m$^2$, and 3.09 kWh/m$^2$ in severe cold, cold and HSCW climate zones, respectively. The results show that energy savings are lowest in cold climate

regions. The reason is that the energy consumption of fresh air has a combined effect on heating and cooling load. Colder winter a hotter summer can both result in larger energy consumption for fresh air. Further, the heat recovery efficiency rate also plays an essential role, and 70% latent heat recovery rate is recommended and adopted.

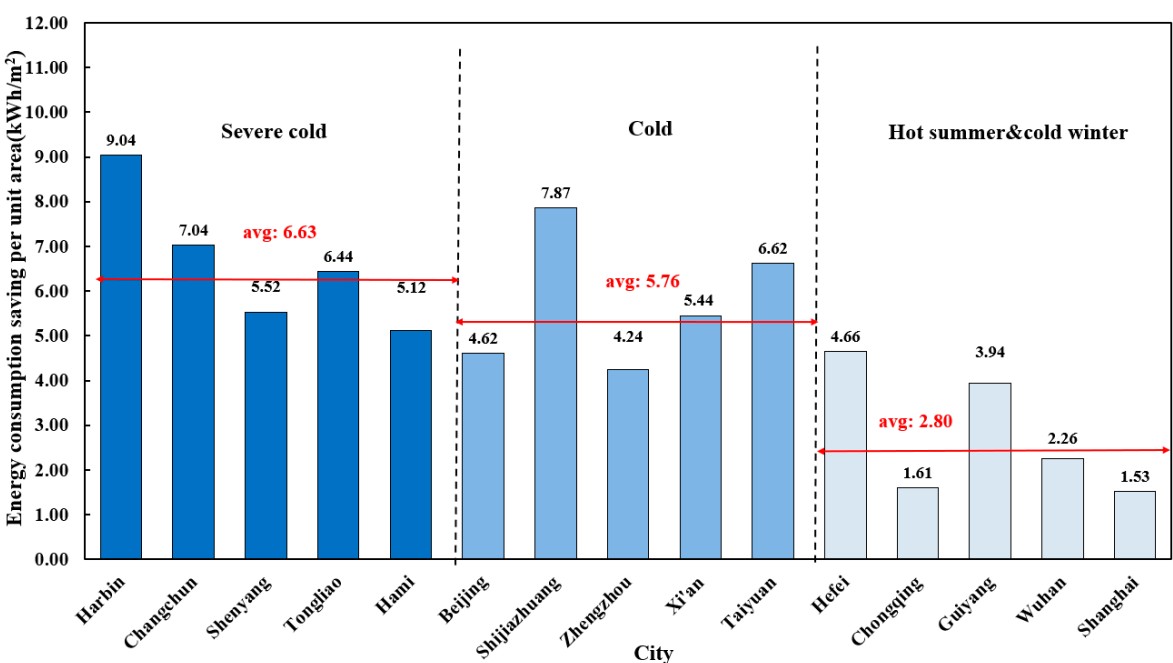

**Figure 5.** Energy saving tendency in severe cold, cold and HSCW climate zones by using high-performance envelopes.

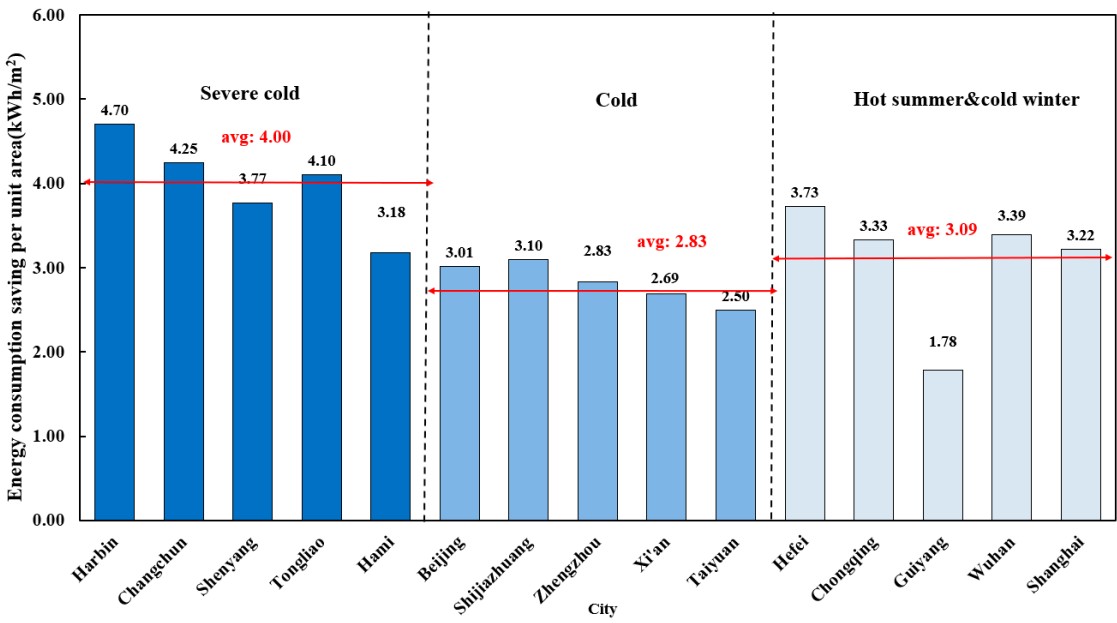

**Figure 6.** Energy saving tendency on severe cold, cold, and HSCW climate zones by using FAHRU.

In addition to the FAHRU, DCV is also considered as the widely used energy saving technology for dealing with fresh air. In European countries such as Norway, DCV is the dominating ventilation strategy [47,48]. This is motivated by the national and EU requirements to reduce greenhouse gasses and profitability in terms of energy savings. Building types with varying occupancies and mechanical system operational periods, such as schools and office buildings, DCV systems can significantly reduce energy consumption.

Figure 7 shows result of average ESPA of DCV indicating that less gap occurred among severe cold, cold, and HSCW climate zones. Nevertheless, more energy reduction could be achieve with DCV, when airflow could vary by the quantities of occupants, together with a combined effect of high COP heating and cooling source and high heat recovery rate unit.

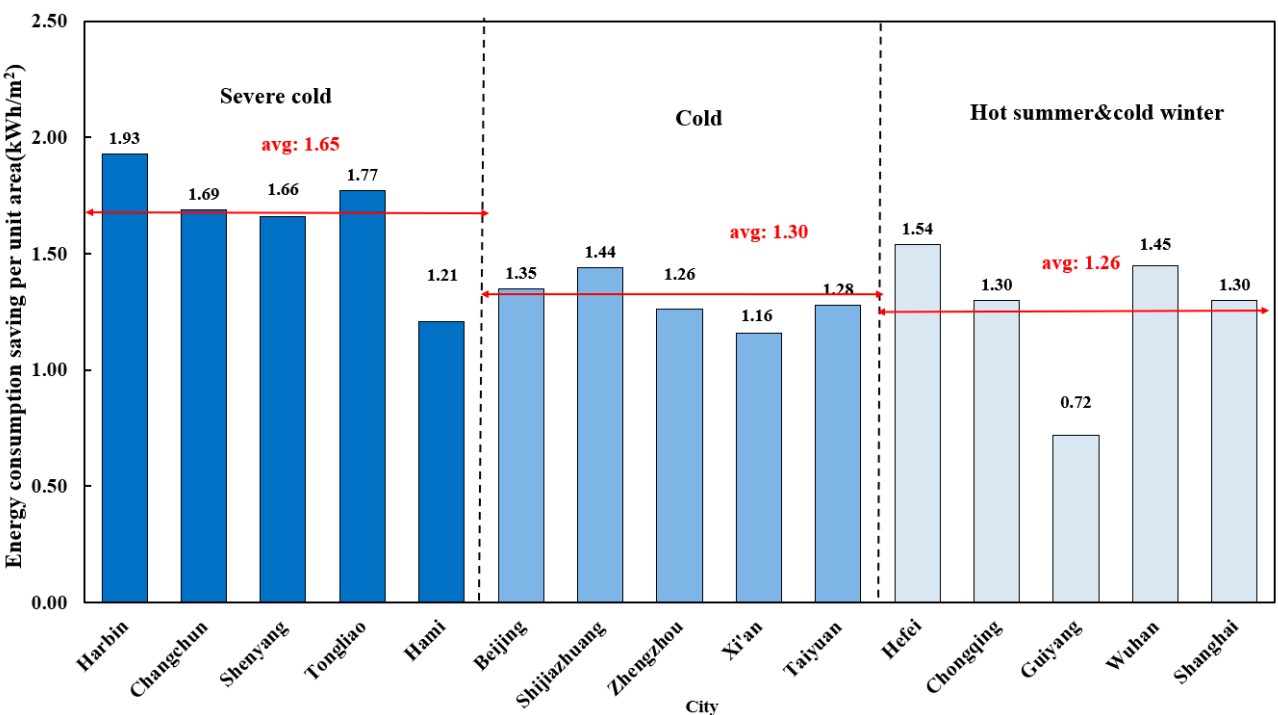

**Figure 7.** Energy saving tendency on severe cold, cold and HSCW climate zones by using DCV.

### 3.1.3. Higher COP of Air Source Heat Pumps

It is well known that higher COP can bring significant energy savings to ordinary buildings. However, it is still unclear of the specific energy saving that can be obtained in an NZEB. Hence, the ESPA in an NZEB is discussed. As presented in Figure 8, averaged energy saving produced by higher COP of air source heat pump contributed more energy consumption in severe cold climate zone and HSCW climate zone. As with the FAHRU, the HVAC operational period and weather conditions are the main impacting factors. Although the average energy saving improvement is 6.43 kWh/m$^2$ in the cold region, it is still contributing significant amount of energy saving.

### 3.1.4. High-Efficient Lighting System

Lighting system provides sufficient visual illuminance for occupants and accounts for a large proportion of energy consumption, approximately one-third of energy consumption in commercial buildings in the U.S. and 20–40% in large office buildings in China [49]. Improving lighting system efficiency and employing effective control strategies are the main optimization methods at present, which are discussed in this section.

In the study case, the lighting power density used in the nearly zero-energy office shows the annual ESPA of 15.81 kWh/m$^2$ with a reduction rate of 56.6% compared with the common office. Due to that, the varied energy consumption can also lead to the change of related HVAC energy consumption. Moreover, to further reduce the energy consumption, daylighting is commonly introduced as a lighting control strategy and the annual ESPA caused by daylighting can achieve 16.4 kWh/m$^2$ with an average window-wall ratio of 40%. The result of integrated impact on building energy saving on lighting and daylighting in different climates has also been illustrated by Ran.W [50].

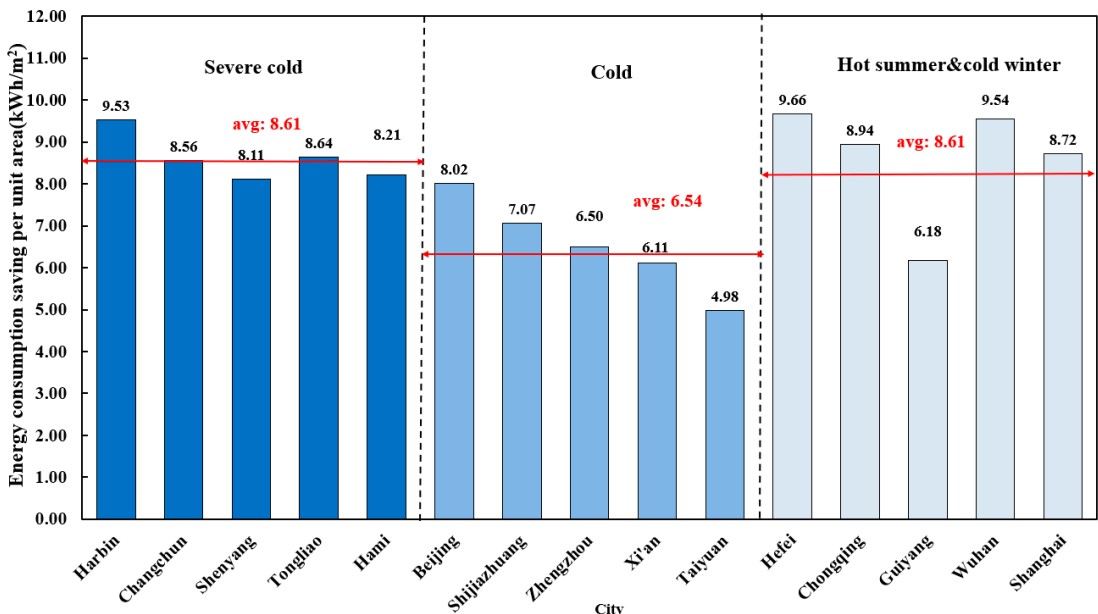

**Figure 8.** Energy saving tendency on severe cold, cold and HSCW climate zones by using COP.

### 3.1.5. PV System

PV is an essential technology to achieve NZEB target [51–53], however, solar energy resources [54], building accepted installed area [55], material types, installed panel angle can affect its electric power generation [56].

In this study, PV SYST has been used to calculate the power generation of the PV system for 15 cities by coordinating the optimized inclination angle to generate maximum electricity. The results of energy saving for PV indicate that the maximum value is 6.35 kWh/m² in Hami and the minimum value is 2.35 kWh/m² in Chongqing. The main reason is that Hami is located in the first-priority region owning the most abundant solar energy with the annual total radiation per unit area higher than 1750 kWh/m² in China. On the contrary, Chongqing is located on the edge of Sichuan Basin which owns the scarcest solar radiation in China [57]. Hence, Figure 9 presents that the cold climate zone has the highest average ESPA of 5.34 kWh/m².

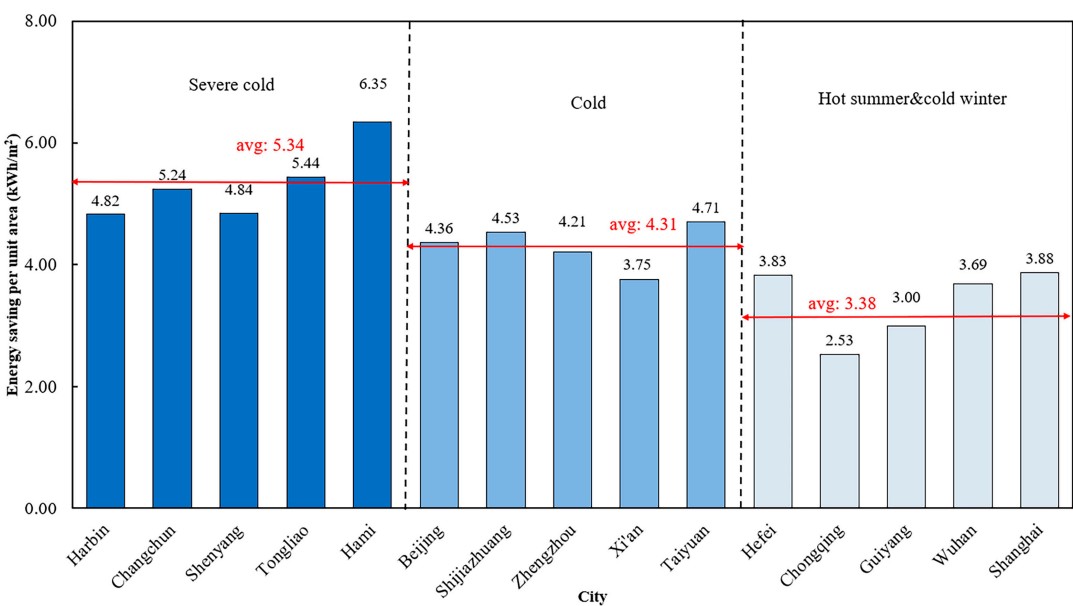

**Figure 9.** Energy saving tendency on severe cold, cold and HSCW climate zones by using PV.

### 3.1.6. Comprehensive Comparison and Discussion

By synthetically considering the energy saving technologies mentioned above, the comprehensive ESPA of the NZEB located in different climate zones can be obtained. As shown in Figure 10, the average annual ESPA in severe cold, cold, and HSCW climate zones are 32.70 kWh/m$^2$, 29.01 kWh/m$^2$, and 27.14 kWh/m$^2$, respectively. It was summarized that the NZEB showcased obvious advantages in different climate zones with considerable energy saving benefits.

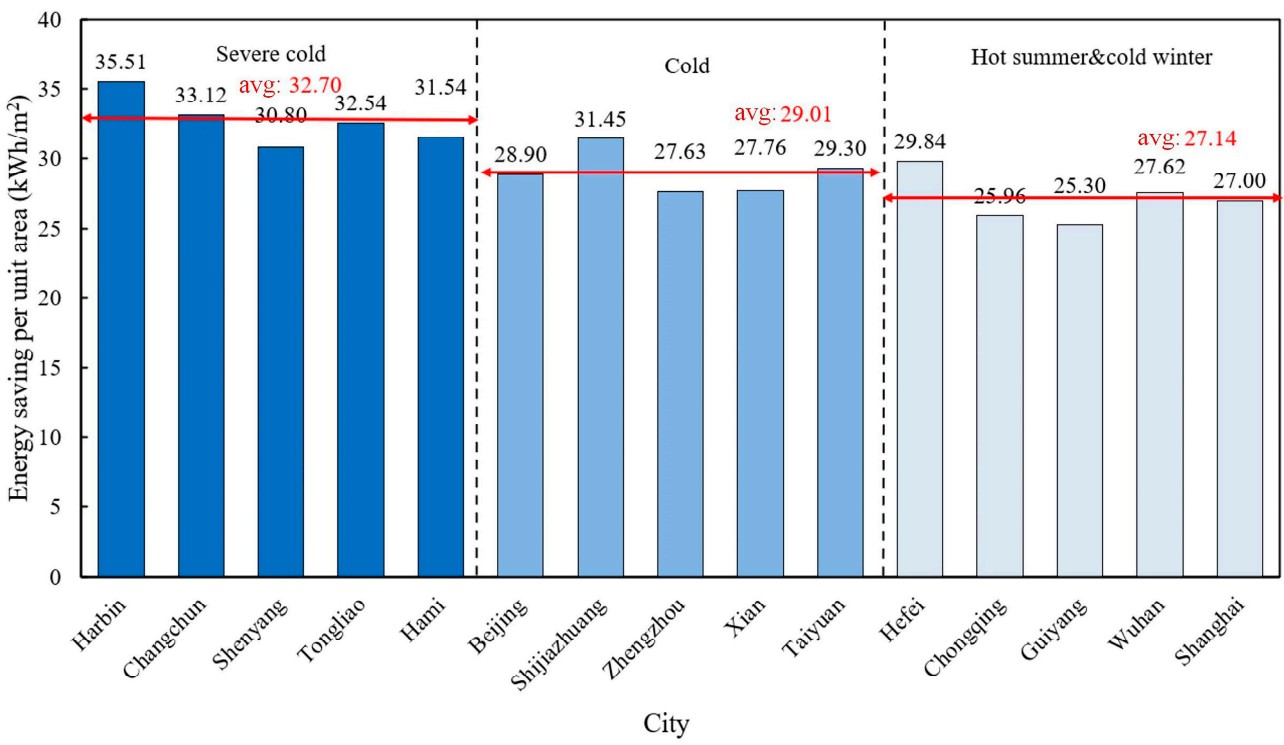

**Figure 10.** Energy saving comparison in different climate zones.

For buildings located in the severe cold climate zone, the energy saving impact ranking is daylighting, high efficiency lighting system, higher COP, high-performance envelopes, PV, FAHRU and DCV. For buildings located in the cold climate zone, the order of the energy saving caused by different technologies from largest to smallest is daylighting, high efficiency lighting, higher COP, high-performance envelopes, PV, FAHRU, and DCV. Hou et al.'s study on building load composition analysis of five different cities in China also confirmed that for Harbin (severe cold climate zone) and Beijing (cold climate zone), the envelope and COP of the HVAC system are the main parameters affecting building energy consumption [58].

For buildings located in the HSCW climate zone, the order of the energy saving caused by different technologies from largest to smallest is daylighting, high efficiency lighting, higher COP, PV, FAHRU, high-performance envelopes, and DCV. For HSCW climate zone, Xu et al. also proved that the building energy saving effect of high-performance envelope was not obvious, while lighting system and high-efficient HVAC system and lighting system contributed more energy saving in HSCW climate zone [59]. It can be concluded that high-efficient lighting system can provide significant contribution to energy saving, as presented in Figure 11.

The above findings of energy saving from analysis of different climate zones may serve as a reference for similar buildings worldwide.

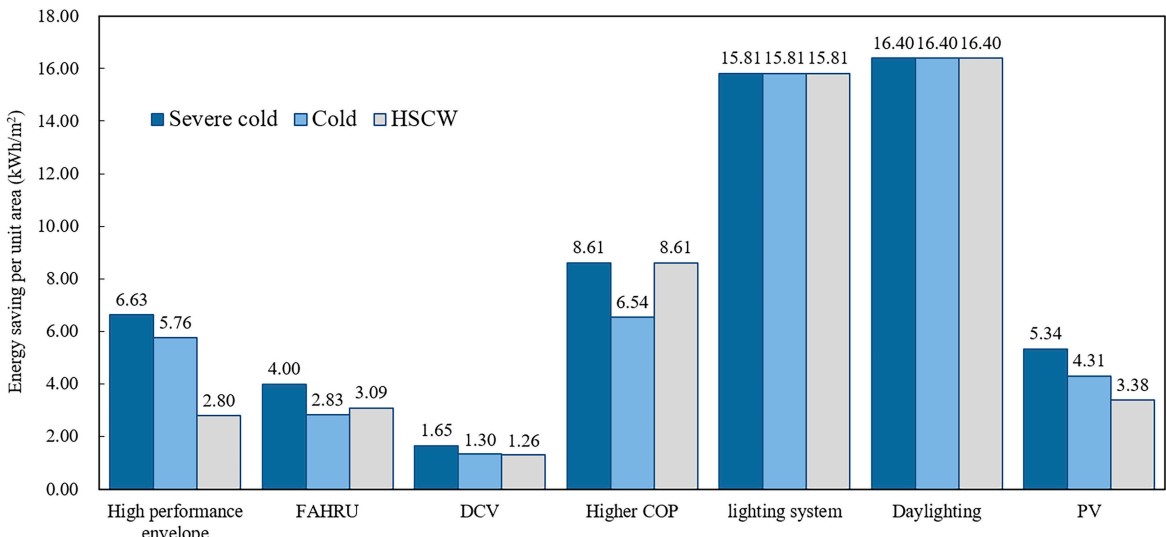

**Figure 11.** Comparison on breakdown of energy saving technologies in different climate zones.

### 3.2. Analysis of Carbon Emission Reduction

In this section, carbon emission reduction amounts and rates of various energy-saving technologies for three climate zones are calculated and compared, as depicted in Figures 12 and 13. The average amount of carbon emission reduction for severe cold, cold, and HSCW climate zone are 21.97 $kgCO_{2eq}/m^2$, 19.60 $kgCO_{2eq}/m^2$, 15.40 $kgCO_{2eq}/m^2$, respectively. HSCW shows the best carbon emission reduction performance due to the integrated reason of low annual energy saving and relatively low average electrical carbon emission factor. Table 5 shows the distribution of electrical carbon emission factors, which is affected by the power generation form [47]. It is indicated that power grids of central China and southern China region have relatively lower carbon emission factors of 0.51 and 0.57 by using more hydro power station and fewer coal-fired power plants.

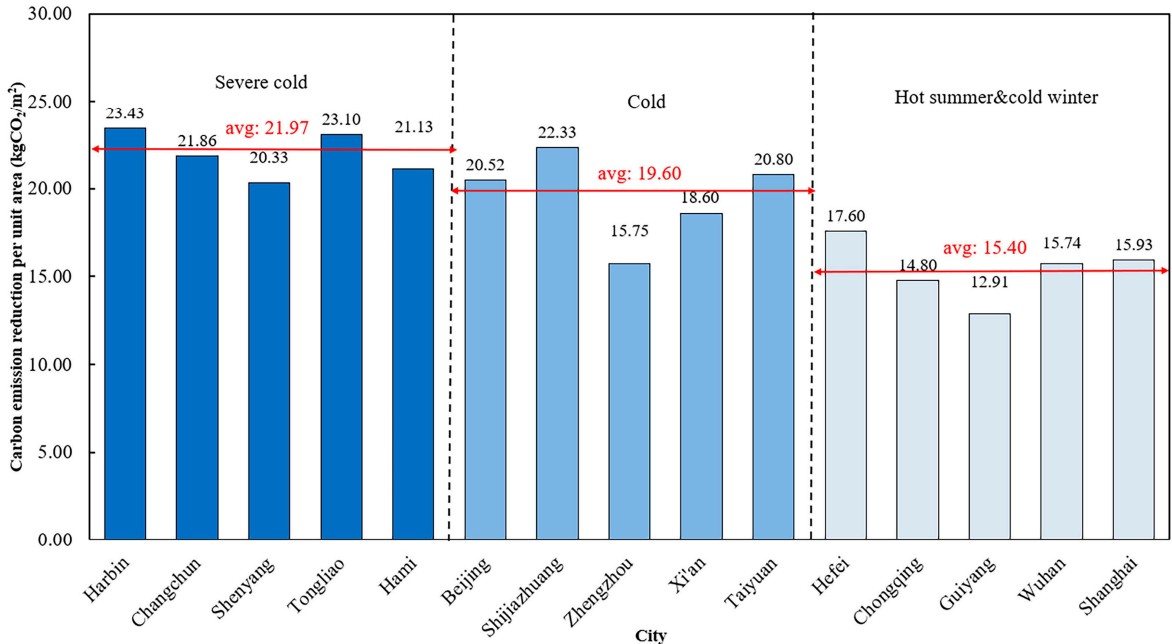

**Figure 12.** Carbon emission reduction per unit area in different climate zones.

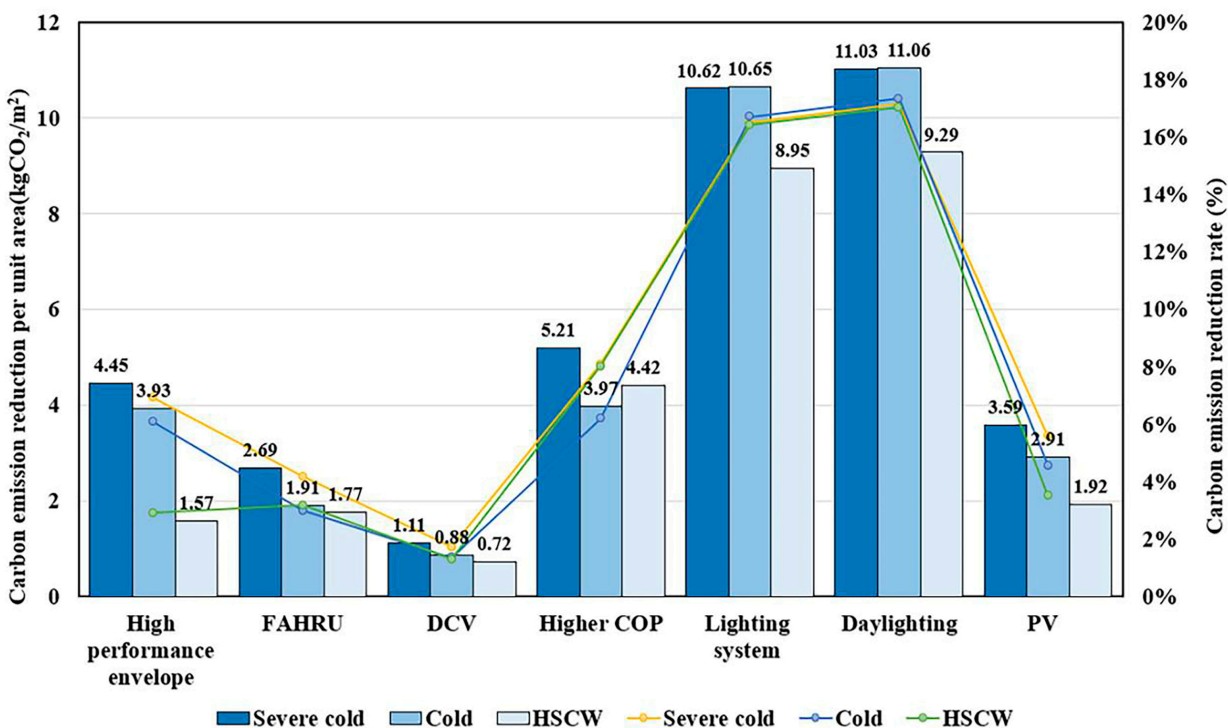

**Figure 13.** Carbon emission reduction amounts and rates of different NZEB strategies.

**Table 5.** Carbon emission factors of China regional power grid.

| China Regional PowerGrid | Typical Cities | $EF_{grid}$ (kgCO$_{2eq}$/kWh) |
|---|---|---|
| Southern Regional Grid | Guiyang | 0.51 |
| Northeast Regional Grid | Harbin; Shenyang; Changchun | 0.66 |
| East China Regional Grid | Hefei; Shanghai | 0.59 |
| Central China Regional Grid | Zhengzhou; Chongqing; Wuhan | 0.57 |
| North China Regional Grid | Beijing; Shijiazhuang; Taiyuan Tongliao | 0.71 |
| Northwest Regional Grid | Xi'an; Hami | 0.67 |

### 3.3. Feasibility of Carbon Emission

To discuss the feasibility of carbon emission, the incremental cost is analyzed. Figure 14 indicates that Harbin has the highest incremental cost of 1017.13 CNY/m$^2$, whereas Guiyang has the lowest value of 612.26 CNY/m$^2$. The incremental cost of severe cold area shows the highest average incremental cost, but HSCW area shows the lowest value. The deviation on average incremental cost of the highest and lowest value are 342.28 CNY/m$^2$, which means more incremental cost are needed in severe cold area to achieve NZEBs. The range of incremental cost is consistent with the results of a previous study in China [25,60].

Figure 15 indicates the incremental costs producing by different energy-saving strategies compared with baseline building and NZEB.

The role of the building envelope is significant in all three climate zones, especially in severe cold area whereas produce medium carbon emission reduction potential. It is a crucial technology and common requirement in building energy saving code to reduce heating load and corresponding primary energy consumption. The performance of envelope could be enhanced by adopting thick insulated materials, triple pane windows with superior airtightness level. In the HSCW region, SC value impacts the carbon emission reduction rate, whereas increasing the K value of envelope is the subordinary factor due to

the weather condition. Adopting of operable external shading system could result in better performance but higher incremental cost.

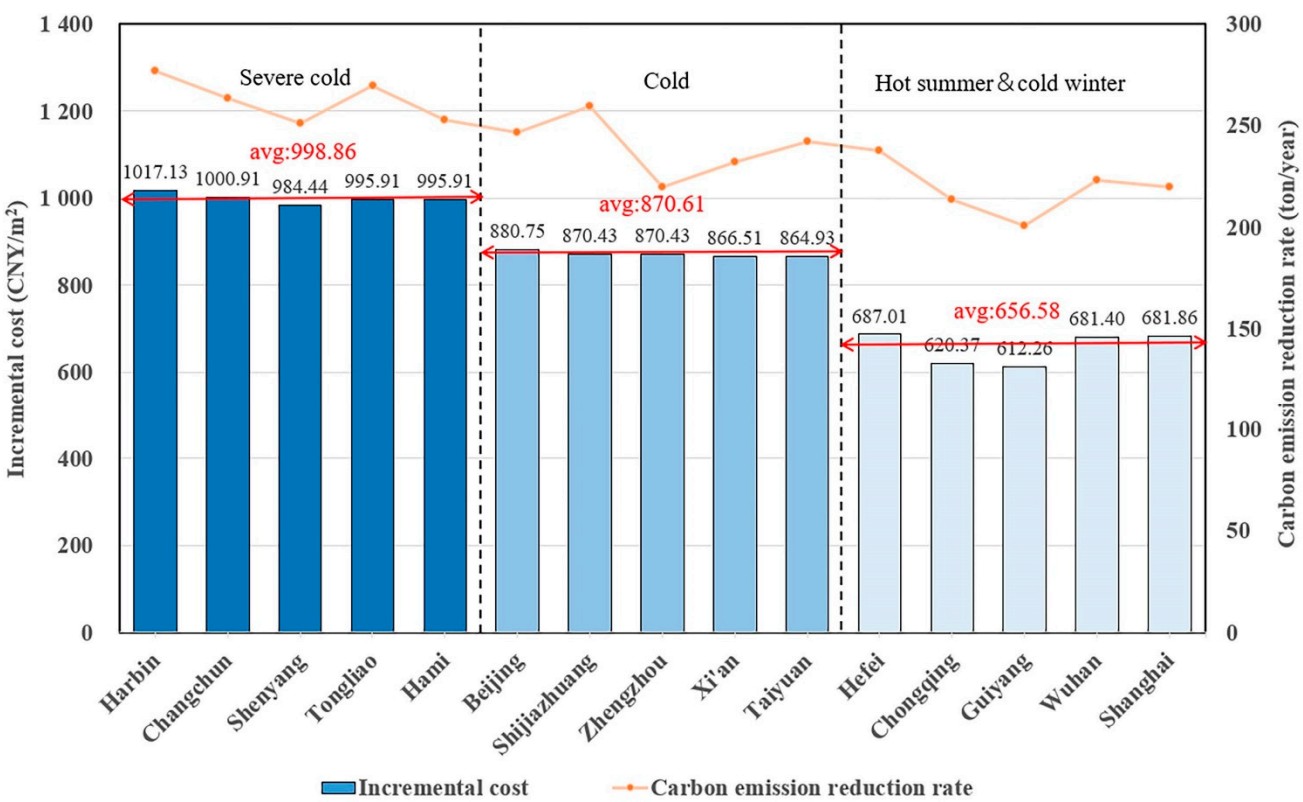

**Figure 14.** Incremental cost and carbon emission comparison in different climate zones.

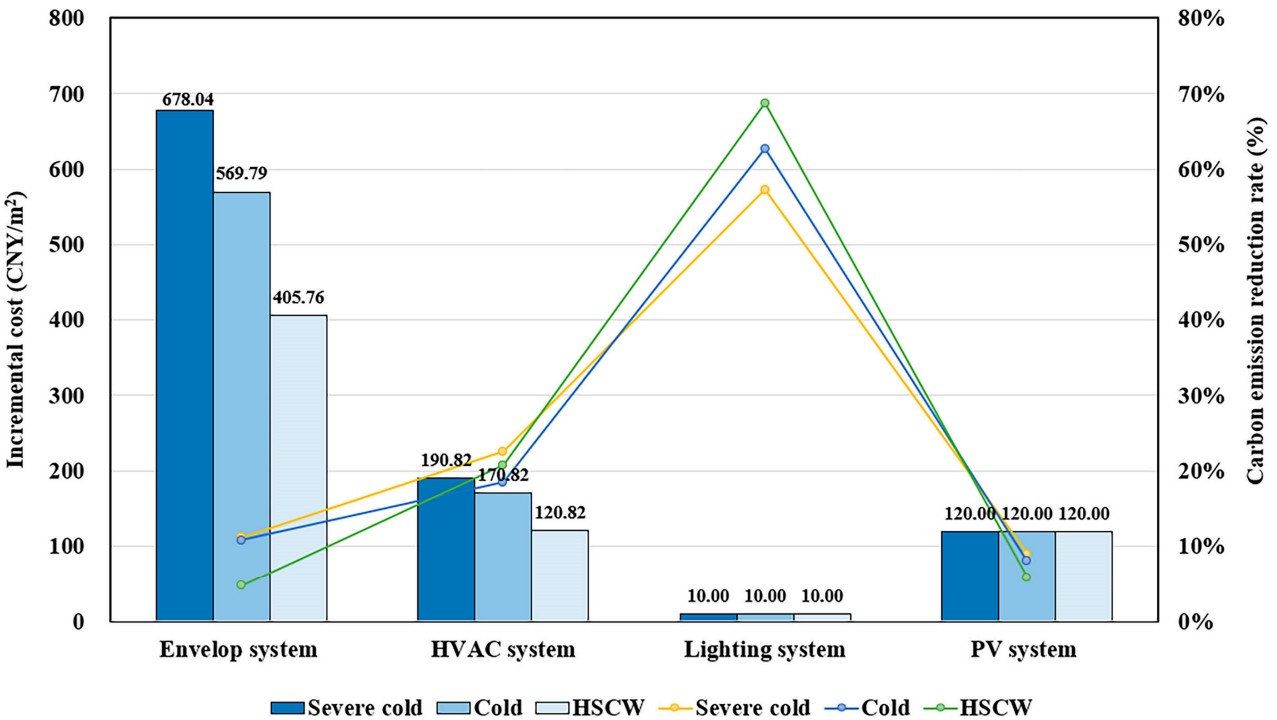

**Figure 15.** Comparison on breakdowns of incremental cost and carbon emission reduction rate in different climate zones.

HVAC system indicates better carbon emission reduction potential effect with relatively low incremental cost especially in HSCW region due to lower heating demand. While high-efficient heating source is needed in severe cold and cold regions. In single view, the incremental cost of HVAC could reduce in NZEB compared to baseline building due to the contribution of low heating and cooling load by using high-performance envelope.

The lighting system shows the best cost performance and high carbon emission reduction rate. The incremental cost of lighting system is not varied by the climate. The incremental costs of PV among three zones are at a relatively consistent level, but severe cold owns a better carbon emission reduction potential rate.

### 3.4. Climate Carbon Emission Sensitivity Analysis

To investigate the effect of building envelope on carbon emissions in three climates zones, 15 cities are selected from the cold zone, severe cold zone, and HSCW zone. As depicted in Figure 16, the lowest and highest amount of carbon emissions are obtained by Guiyang and Shijiazhuang, corresponding to a value of 303.8 t and 456.7 t, respectively. The average carbon emission in severe cold zone was the highest, and that in HSCW climate zone was the lowest. However, based on the analysis of energy consumption in the three climate zones, building energy consumption in HSCW climate zone is the highest, and the reason for its low carbon emission is that the value of the carbon emission factor of the power grid in this climate zone is generally lower than that of the other two climate zones. The average carbon emission in the HSCW zone is the lowest as heat transfer coefficients of building envelopes are the lowest, while that in the cold zone is the highest, as its meteorological conditions and insulation performance of building envelopes are at the middle level. Based on the analysis of the NZEB technology, it can be found that after improving the thermal performance of the building envelope, the carbon emission can be reduced by 1.6–9.3%, with an average reduction of 5.3%. Particularly, the average carbon emissions reduction rate in the cold zone, severe cold zone, and HSCW zone are 6.1%, 6.9%, and 2.9%, respectively, indicating that improving building envelops in severe cold and cold areas are more helpful to decrease carbon emissions.

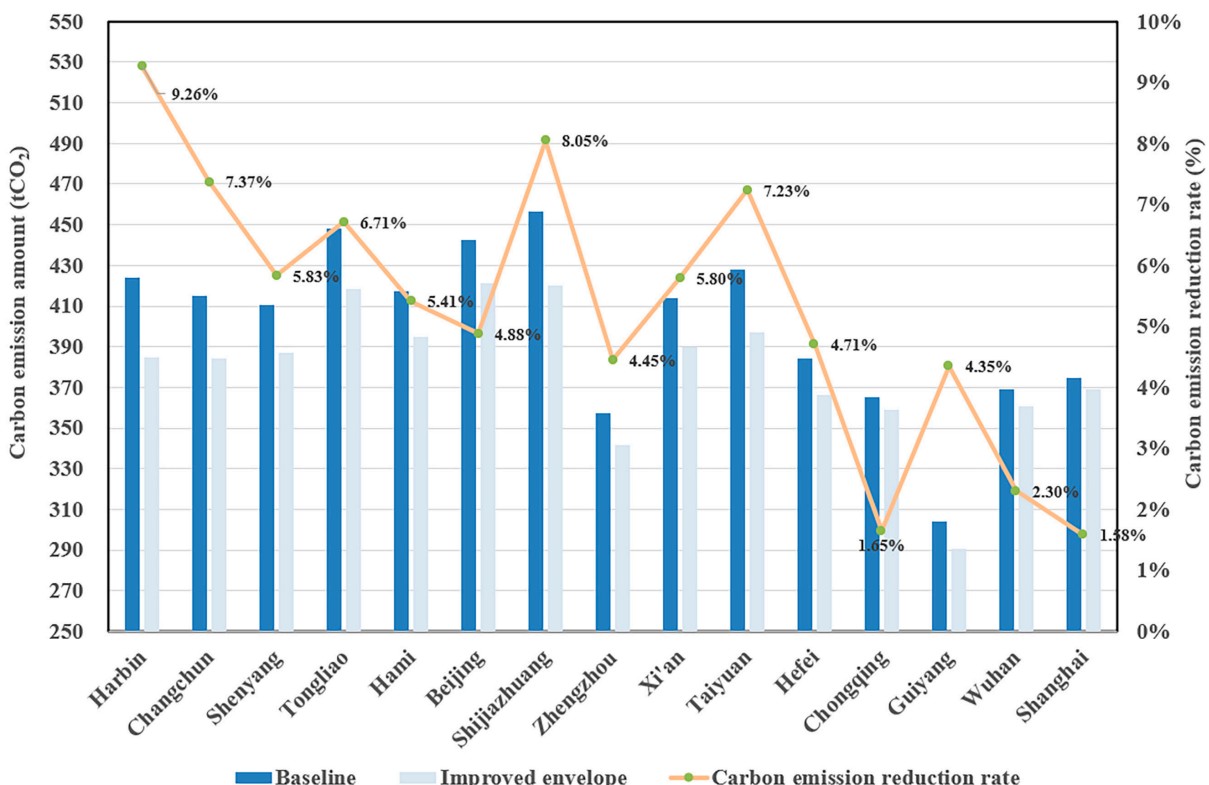

**Figure 16.** Carbon emissions amount and reduction rates after improving building envelope.

When intervening FAHRU, carbon emission amounts can reduce by 6.0–20.4 t. The highest carbon reduction rate is obtained by Harbin (located in severe cold zone) with a value of 4.8%, and the lowest carbon reduction rate is obtained by Guiyang (located in HSCW zone) with a value of 2.0% (shown in Figure 17). Among these three zones, the averaged carbon emissions reduction rate in the cold zone is the lowest, as its meteorological conditions are at the middle level, and operating time for heating and cooling are relatively shorter. On the other hand, if DCV is added for indoor fresh air volume control, less fresh air will be required, and carbon emissions can be decreased by 0.8–2.0%. As a result, it is beneficial to further considering the combination of FAHRU with DCV to further improve the energy saving and carbon emission reduction.

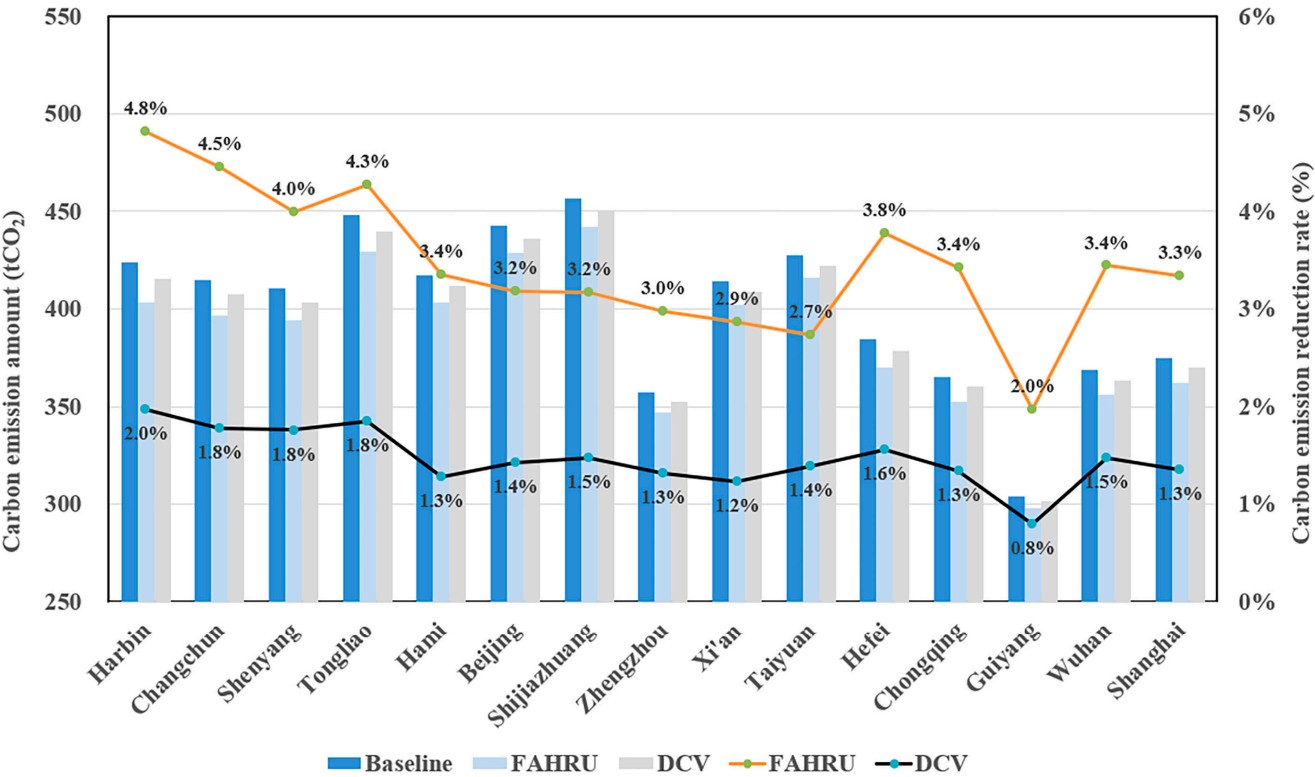

**Figure 17.** Carbon emissions amount and reduction rates after using FAHRU and DCV.

COP influences the output of air source heat pumps, thus affecting the energy consumption and carbon emissions caused by heating and cooling load. When the COP changes between 3.18–2.28, carbon emission amounts increase by 18.7–37.3 t $CO_2$ respectively, as presented in Figure 18. The carbon emission rates increase in the cold zone, severe cold zone, and HSCW zone are 6.2%, 8.1%, and 8.0%, respectively, denoting that HVAC systems with superior performance are urgently required in NZEBs.

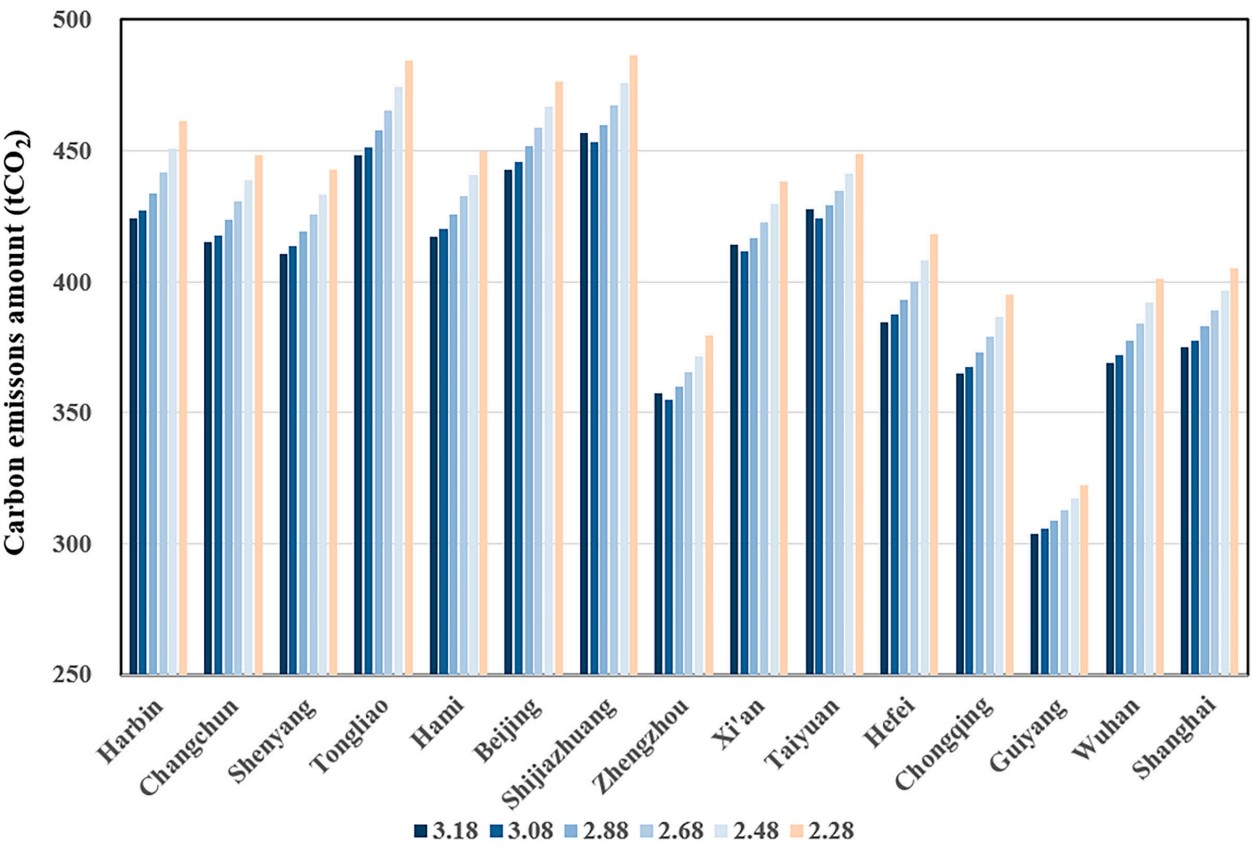

**Figure 18.** Carbon emissions amount when adopting air source heat pumps with different COP value.

## 4. Conclusions

This paper studied a comprehensive and multiple climatical carbon emission and cost analysis based on a NZEB case. Various energy-saving technologies were adopted and analyzed for 15 typical cities in severe cold, cold, and HSCW climate zones of China.

By using NZEB technologies, the annual aggregated yearly carbon emission and categorized carbon emission for NZEB technologies are analyzed and discussed in three climate regions. Buildings in the HSCW region may have the lowest carbon emission reduction potential through the comprehensive effect on energy saving potential and lower electricity carbon emission factors. By introducing an analysis of incremental cost, lighting system and daylighting show the best carbon emission reduction potential and produce the lowest incremental cost.

In the view of climatic zones, buildings in severe cold and cold climate have a high incremental cost of 998.86 CNY/m$^2$ and 870.61 CNY/m$^2$. Buildings in HSCW show the best carbon emission reduction potential and cost effectiveness by adopting NZEB strategies. It is found that the comprehensive strategies of improving heat transfer performance of envelope and promoting efficient HVAC system show a greater carbon emission reduction potential for NZEB in severe cold and cold regions than in HSCW regions. Although incremental cost of passive strategies produced by envelop system is higher than active strategies produced by HVAC system and lighting system, the effect of reducing building heating load is primary and necessary. Further, more strategies of reducing carbon emission such as carbon offset and green power would be other essential pathways for the development of clean energy gird and building electrification.

The research outcome of this study may provide valuable references for aspects of calculation method, sensitive analysis, and incremental cost for similar types of buildings in different regions worldwide.

**Author Contributions:** Y.K.: Conceptualization, methodology, formal analysis, writing-original draft, data curation, validation, visualization. J.W.: methodology, formal analysis, supervision. S.L.: conceptualization, investigation, writing-review and editing. Y.Y.: software, writing-review and editing. Z.Y.: software, writing-review and editing. H.Z.: writing-review and editing. S.X.: writing-review and editing. Z.F.: data curation, validation. M.F.: writing-review and editing. X.X.: writing-review and editing. All authors have read and agreed to the published version of the manuscript.

**Funding:** This work was funded by the CABR-IBEE Corporate Research Fund-Nearly Zero Energy Building Performance Monitoring and Continuous Commissioning [grant number 20200303310730001] and CABR-IBEE Corporate Research Fund-Study and Development of High Performance Building Intelligent System and Nearly Zero Energy Building Operation and Management System [grant number 20200303310730002].

**Institutional Review Board Statement:** Not applicable.

**Informed Consent Statement:** Not applicable.

**Data Availability Statement:** Not applicable.

**Acknowledgments:** This work was supported by the CABR Corporate Research Funds.

**Conflicts of Interest:** The authors declare no conflict of interest.

## Nomenclature

| | |
|---|---|
| $A$ | Building construction area |
| $\alpha$ | Annual energy saving per area |
| $\beta$ | Annual carbon emission reduction per area |
| $\gamma$ | Annual incremental cost per ton per area |
| $C1$ | Original cost without energy-saving technologies |
| $C2$ | Cost by using NZEB technologies |
| $CE1$ | Original carbon emission without energy-saving technologies |
| $CE2$ | Carbon emission by using NZEB technologies |
| $E_1$ | The original energy consumption without energy-saving technologies |
| $E_2$ | The energy consumption, introducing corresponding energy-saving technology |
| $EFgrid$, | Carbon emission factor of the grid in the local area |
| $R$ | Carbon emission reduction rate |

## Abbreviations

| | |
|---|---|
| APEC | Asia-Pacific Economic Cooperation |
| CDD | Cooling degree days |
| COP | Coefficient of performance |
| DCV | Demand-control ventilation |
| EPS | EnergyPlus software |
| ESPA | Energy saving per area |
| EU | European Union |
| FAHRU | Fresh air heat recovery unit |
| HDD | Heating degree days |
| HVAC | Heating, ventilation, and air conditioning system |
| IPCC | Intergovernmental Panel on Climate Change |
| NZEB | Nearly zero-energy building |
| PV | Photovoltaic |
| SC | Shading coefficient |
| SHGC | Solar heat gain coefficient |

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
