# Peer review of "Comprehensive Carbon Emission and Economic Analysis on Nearly Zero-Energy Buildings in Different Regions of China"

_sustainability, doi:10.3390/su14169834_

Round 1

Reviewer 1 Report

Recommendation to the authors:

The topic of the paper entitled “Comprehensive carbon emission and economic analysis on nearly zero-energy buildings in different regions of China” is interesting and worthy of investigation. Also, this paper is well written, including introduction, methodology, results, and conclusions, however, I cannot accept this paper in its current state. Below are some recommendations that the reviewer thinks will improve a future version of this work.

- Abstract: The authors should improve the last part of the abstract “The simulated carbon emission intensity in severe cold, cold, hot summer and cold winter (HSCW) climate zones ……The findings may provide reference for similar buildings in different climate zones worldwide.”, to include 2-3 key findings of this study; and key conclusion.

- Introduction: the originality and contributions of this study should be clearly addressed. The aim of this study is insufficient to convince readers and researchers worldwide. 3-4 sentences, please, should be added to the end of the introduction to address this issue.

- Methodology: 1) Figure “(a) Typical cities on climate zones”, please add legend to the map.  2) Figures 3 and 4, should be improved.  3) “2.1.3. Carbon emission reduction model”, this subsection should be improved with clear equations, for example, why not use “g CO2eq/kWh” instead of “carbon emission factor (tCO2/MWh) “to explain how much can NZEBs reduce from carbon per kWh.  4) Please also clearly state which NZEB scenarios are included in the paper and the specific characteristics of each scenario. 5) Please check again the tables 2 and 3.

- Results: 1) Figures 5-8, should be improved with a clear font. 2) Why use (kgCO2/m2) and (tCO2)? It is better to use a standard unit along the paper/ manuscript. Also, please improve the quality of some subsections by focusing on key findings.

- Conclusions: Limitations of the study and potential future development of this research should also be included in the manuscript.

Please include a point-by-point reply to the above comments. Please detail the revisions that have been made, citing the line numbers and the changes made.

Reviewer 2 Report

It is an article that deals with a topical issue, but that is already evolving: nearly zero energy building. But current research is already beginning to consider buildings.  Zero Net Energy and Carbon in Buildings and Positive Energy Buildings and Carbon free. This reflection on the future state of the building is absent in the article.

It necessary in the calibration of the simulated model with the actual data, values the percentage of error to give it as valid. Missing references to support error percentage decision

It would be interesting to indicate in point 3.2, why for the same energy source the passing factors of the same energy are different according to the area of the country

The conclusions are insufficient. Most of the text is intended to include the wording of the article. The relationship between nzeb and active and passive strategies of conditioning, the cost-effectiveness of different combinations of strategies, the relationship between region and type of strategy of channeling (active or passive) and its efficiency, etc.

Reviewer 3 Report

The manuscript deals with the important topic of reducing carbon emissions by using the nearly zero-energy buildings in the building sector based on four strategies. The idea is interesting and worthwhile investigating. But there are drawbacks in some sections of the manuscript. The knowledge gaps should be clearly identified and were required to link to your paper goals.

Major comment

The problem background needs alignment with the introduction and literature sections because the proper comparison is not there and not clear; it needs just why the existing studies in NZEB are not able to show the reducing carbon emission, energy-saving and thermal comfort. Furthermore, the article lacks a significant criticism in the review element which is related to characteristics of different climate zones, there are many works that investigated the right choice for the zone location saving more than 50 % of HVAC system energy, such as. https://doi.org/10.1016/j.scs.2020.102091

 Improving carbon emission reduction is essential for sustainable development; so, this study is needed to assess the previous studies contribute to the enhancement of the (1) passive design and building envelope; (2) high-performance heating, ventilation and air conditioning (HVAC) system; (3) high-efficient lighting system; and (4) renewable energy utilization.

Minor comment

a. Please in subsection 2.2.2 needs to show the physical descriptions for the model of case building.

b. English is ok. However, some typos must be checked.

c. In subsection 2.1.3 needs to explain scheduling the lights to come on at the right time-based lighting control such as. https://doi.org/10.1016/j.apenergy.2022.118863

d. The authors need to explain in detail the passive design and building envelope.

e. Figures are ok and they help in the manuscript's description.

f. In Table 2. Indoor parameters did not follow ASHRAE Standard 55–2010 and ISO 7730.

g. What about the limitations of the study? Please, improve the revised version of the manuscript.

h. The conclusion needs to be well presented for the overall understanding of the research.

Round 2

Reviewer 1 Report

Compared to the previous draft, the authors have addressed some of the reviewers' comments.

Reviewer 2 Report

Congratulations for the work done. Thank you very much for responding to the initial indications of the review

Reviewer 3 Report

The authors have addressed my comments and I am satisfied with the improvements.